# A Novel Cross-Sensor Transfer Diagnosis Method with Local Attention Mechanism: Applied in a Reciprocating Pump

**DOI:** 10.3390/s23177432

**Published:** 2023-08-25

**Authors:** Chen Wang, Ling Chen, Yongfa Zhang, Liming Zhang, Tian Tan

**Affiliations:** 1School of Nuclear Science and Technology, Naval University of Engineering, Wuhan 430033, China; m22382701@nue.edu.cn (C.W.); goodcl@126.com (L.C.); zlm060101@aliyun.com (L.Z.); tantian960124@163.com (T.T.); 2Chongqing Pump Industry Co., Ltd., Chongqing 400033, China; 3Chongqing Machine Tool Co., Ltd., Chongqing 401336, China

**Keywords:** reciprocating pump, fault diagnosis, transfer learning, local attention mechanism

## Abstract

Data-driven mechanical fault diagnosis has been successfully developed in recent years, and the task of training and testing data from the same distribution has been well-solved. However, for some large machines with complex mechanical structures, such as reciprocating pumps, it is often not possible to obtain data from specific sensor locations. When the sensor position is changed, the distribution of the features of the signal data also changes and the fault diagnosis problem becomes more complicated. In this paper, a cross-sensor transfer diagnosis method is proposed, which utilizes the sharing of information collected by sensors between different locations of the machine to complete a more accurate and comprehensive fault diagnosis. To enhance the model’s perception ability towards the critical part of the fault signal, the local attention mechanism is embedded into the proposed method. Finally, the proposed method is validated by applying it to experimentally acquired vibration signal data of reciprocating pumps. Excellent performance is demonstrated in terms of fault diagnosis accuracy and sensor generalization capability. The transferability of practical industrial faults among different sensors is confirmed.

## 1. Introduction

Reciprocating pumps are widely used in critical industrial sectors, such as petroleum and chemical engineering, particularly offshore oil production, operating in high-temperature, high-pressure, and corrosive environments. Due to the harsh offshore working conditions, maintenance, fault detection, and repairs of reciprocating pumps are challenging [1]. The occurrence of a failure can lead to significant disasters. Therefore, effective health monitoring and intelligent diagnosis technology are essential to enhance industrial production efficiency.

In fault diagnosis for reciprocating pumps, selecting appropriate monitoring techniques is paramount to ensure efficient and accurate detection and diagnosis of potential issues within the system. Methods, such as orbit shape analysis, deflection shape analysis, and acoustic emission analysis, can also provide valuable insights into specific scenarios, but they may exhibit limitations in certain aspects. For instance, orbit shape analysis [2] can be influenced by the intricate geometric complexity of reciprocating pumps; deflection shape analysis [3] is constrained by requirements for high-precision measurement equipment, susceptibility to environmental interference, and intricate data processing challenges. Kumar et al. [4] proposed a modified CNN to identify the common faults of a centrifugal pump based on acoustic signal, including bearing with the inner and outer race defect and clogging in the impeller and a broken impeller. However, this acoustic emission analysis is susceptible to significant interference from environmental noise factors [5]. Tang et al. [6] constructed an adaptive convolutional neural network (CNN) using the pressure signal of the pump. It can effectively identify different fault patterns of hydraulic piston pumps. The pressure analysis of reciprocating pumps is relatively insensitive to capturing specific mechanical fault patterns, such as bearing faults or liquid leakage. It is challenging to distinguish various types of faults accurately. In contrast, vibration signal analysis methods have found widespread application in industrial environments, possessing unique advantages in mechanical systems. Vibration signals can capture subtle motion variations and structural oscillations associated with a range of mechanical faults, including but not limited to bearing wear, imbalance, and mechanical looseness, thereby providing rich information for fault analysis. Furthermore, a substantial body of prior research has demonstrated the successful application of vibration signals in similar systems. Ahmad et al. [7] proposed a fault diagnosis method for multistage centrifugal pumps using informative ratio principal component analysis (Ir-PCA); the method selected the fault-specific frequency band from the raw vibration signal, and the combination of an informative ratio-based feature assessment and principal component analysis forms the novel informative ratio principal component analysis. Tang et al. [8] introduced a deep learning approach combined with Bayesian optimization to achieve intelligent fault recognition in hydraulic piston pumps. They utilized time-frequency images obtained through continuous wavelet transform of vibration signals as input data. Ahmad et al. [9] proposed a novel framework for centrifugal pump fault diagnosis based on selective Walsh transform and cosine linear discriminant analysis of fault characteristic coefficients by utilizing the vibration characteristics of a centrifugal pump with soft mechanical faults. Zhao et al. [10] proposed an adaptive Variational Time-Domain Decomposition (VTDD) method for identifying multiple impact vibration signals in reciprocating machinery. They effectively identified faults by leveraging the concentrated energy distribution and rapid amplitude variation characteristics of impact signals, highlighting their significant potential in reciprocating machinery fault diagnosis. Hence, the present study aims to explore vibration further in signal-based intelligent fault diagnosis methods for reciprocating pumps, enabling an accurate assessment of the system’s health state.

With the rapid advancement of computer technology, data-driven approaches have been extensively employed in the intelligent fault diagnosis of machinery [11,12,13]. This objective is to eliminate the reliance on manual feature extraction and expert knowledge by directly extracting features from raw data. Then, the successful constructions of data-driven models still heavily depend on ample-labeled training samples [14,15]. Furthermore, the test and training samples must follow the same distribution to obtain reliable diagnostic results. Most existing fault diagnosis studies assume that data collection occurs at the same location for each machine [16]. However, in real-world industrial applications, it is often challenging to fulfill this assumption, particularly for large mechanical equipment, like reciprocating pumps. Due to reciprocating pumps’ unique structural conditions and operational environments, the available space for installing vibration sensors on the pump head position is limited and inconvenient to adjust. In particular, for long-term deployments, cable management of sensors can be compromised, potentially leading to hazards. Consequently, vibration sensors are difficult to install in the most sensitive pump head positions that reflect faults. Instead, they are typically installed in the machine base positions that offer adequate available space and ease of management. However, vibration signals and noise are transmitted through the vibration source to the machine foot, where they are captured by sensors, making it challenging to obtain the pumps’ fault signals accurately. When the sensor location changes, the feature distribution of the signal data also changes. It can be said that this poses a challenge to the fault diagnosis of reciprocating pumps. In practical engineering applications, the variability in working conditions leads to inconsistent data distributions, making methods based on transfer learning highly favored in addressing cross-domain machinery condition-monitoring problems [17,18,19]. Generally, the generalization ability of transfer learning across different scenarios is improved by transferring knowledge learned from the source domain (training data) to the target domain (testing data). Therefore, in this study, we explore the use of transfer learning to assist the monitoring task of other position sensors using the information already collected by sensors at a particular location, thereby achieving a more accurate and comprehensive fault diagnosis.

In addition, attention mechanisms have been widely applied in fault diagnosis in recent years [20]. Attention mechanisms significantly enhance the intelligent fault diagnosis model’s ability to process mechanical monitoring signals, such as vibration signals and equipment images, and have shown improved application effectiveness in various research, for example, complex equipment, engines, bearings, gearboxes [21], etc. However, several mainstream attention mechanisms in the current field of fault diagnosis have some limitations. For instance, channel attention weights different channels of vibration signals based on their relationships but overlooks the positional relationships of the signals in the time series. This leads to the inability to accurately capture the temporal variations of fault signals [22]. Self-attention mechanisms are more suitable for processing shorter sequences but may encounter issues, such as imbalanced information propagation or gradient vanishing, when dealing with longer sequences [23]. Global attention disperses attention weights excessively, allocating excessive attention to irrelevant parts of vibration signals [24]. Inspired by the ideas of local attention mechanisms in neural machine translation [25] and speech recognition [26], we introduce the concept of a local attention mechanism to the field of fault diagnosis.

The local attention mechanism focuses on specific regions of the input information, shifting from a global perspective to a local one. By applying adaptive weights, it highlights the features of local regions, allowing for better identification of fault characteristics when dealing with long sequences, like vibration signals. Unlike global attention, with the integration of local attention, the contributions made by each position are more pronounced, while important contextual information is still preserved. Therefore, it is more suitable for addressing problems in the domain of the time series, such as speech and signal processing.

Furthermore, to the best of the authors’ knowledge, the current research on fault diagnosis mainly focuses on small-scale mechanical components, such as bearings [27,28,29] and gears [30], rarely lacking in-depth exploration of fault experiments on large-scale machinery equipment, such as reciprocating pumps. Compared to small-scale mechanical components, reciprocating pumps, as typical large-scale machines, exhibit a certain level of complexity and uniqueness in their fault patterns and characteristics. By conducting actual fault experiments on reciprocating pumps, we aim to bridge this knowledge gap in the existing research and provide a more practical insight into the fault diagnosis of large-scale machinery equipment.

To address the issues above, this paper proposes a cross-sensor domain transfer diagnostic method based on local attention. Specifically, we introduce the local attention mechanism into a convolutional model, shifting the focus from the entire vibration signal to local regions. By applying adaptive weights, the features of critical regions are highlighted, enhancing the model’s perception of fault signals’ essential parts. Furthermore, to tackle the challenges of sensor installation difficulties at the pump head, high levels of noise interference, and poor signal quality from sensors at the machine foot position in reciprocating pumps, we present a cross-sensor domain transfer diagnostic method from the pump head to the machine foot. Finally, we conduct experiments targeting the common faults in reciprocating pump valve assemblies in the industry. The proposed method is validated by applying it to experimentally acquired vibration signal data of reciprocating pumps. Excellent performance is demonstrated in terms of fault diagnosis accuracy and sensor generalization capability. The transferability of practical industrial faults between the pump head and the machine foot is confirmed through the validation process. The main contributions of this study are as follows:A cross-sensor transfer diagnostic approach is proposed, which utilizes the sharing of information collected by sensors between different locations of the machine, achieving a more accurate and comprehensive fault diagnosis of reciprocating pumps.A local attention mechanism is embedded in the proposed approach and applied in fields of intelligent data-driven fault diagnosis to enhance the model’s perception of the critical part of the fault signal.Experimental tests on fault samples of a reciprocating pump demonstrate the excellent performance of the method in terms of fault diagnosis accuracy and sensor generalization ability, validating the cross-sensor domain transferability in practical industrial reciprocating pump faults.

## 2. Related Works

### 2.1. Transfer Learning

The typical fault diagnosis problem has been satisfactorily resolved through data-driven methods [31]. However, for more practical engineering applications, the varying working conditions lead to inconsistent data distributions, and transfer learning for fault diagnosis is highly favored for wide applications [16]. In general, the model’s generalization ability in different scenarios is improved by transferring the knowledge learned from the source domain (training data) to the target domain (testing data) [32,33]. Existing research on transfer learning for a diagnosis mainly focuses on fault classification under different working conditions [34,35], across different machines [36,37], and dealing with imbalanced instances [38,39], achieving a good cross-domain diagnostic performance. Yang et al. [14] proposed a CNN-based approach to perform transfer diagnosis tasks from a laboratory-bearing dataset to a locomotive-bearing dataset. Wen et al. [40] designed a transfer learning model based on sparse autoencoders to predict bearing fault types under different operating conditions. Yang et al. [41] presented an unsupervised feature extraction and transfer learning-combined fault recognition method applicable to sucker rod pump faults under different working conditions.

However, most transfer studies assume that sensor data collection is performed at the same location for each machine. In contrast, domain adaptation across different sensors has received much less attention in the current literature [42]. It is worth noting that this assumption is often challenging to achieve in real-world environments. The source domain (training data) and target domain distribution (testing data) may vary with the sensors or their positions. Scenarios involving collecting training and testing data from different locations are rarely considered. To overcome this issue, Pandhare et al. [43] utilized CNN and Maximum Mean Discrepancy (MMD) for fault diagnosis of different sensor positions in ball screw drives. Chen et al. [44] proposed a transformer-based cross-sensor domain fault diagnosis method for aerospace electromechanical actuators. Zhang et al. [45] proposed cross-domain discriminative subspace learning (CDSL) for migration recognition across multiple systems. Se et al. [46] present a novel drift compensation framework, CSBD-CAELM, that integrates cross-domain subspace learning and balanced distribution to achieve dual drift compensation at both feature and classifier levels for gas sensors. Li et al. [47] introduced an adversarial training approach for transfer fault diagnosis in cross-sensor domains, where data collected from different sensors are projected into a shared subspace. Despite the prevalence of cross-sensor domain fault diagnosis in practical industrial scenarios, research on this topic still needs to be explored.

### 2.2. Local Attention Mechanism

In recent years, attention mechanisms have become a hot topic in deep learning, extensively studied and applied by researchers in natural language processing [48] and computer vision [49]. In intelligent fault diagnosis for machinery, attention mechanisms are crucial for capturing internal correlations and enhancing information extraction capabilities. Recently, attention mechanisms have gained popularity in mechanical fault diagnosis, becoming an important technology researched and applied by scholars [20]. Attention mechanisms significantly improve the model’s ability to process mechanical monitoring signals, such as vibration signals and equipment images, and demonstrate performance improvements in various research objects, including complex machinery, engines, bearings, and gearboxes [21]. Specifically, spatial attention is primarily used for mechanical fault classification, aiding CNNs [50] to expand their perception field and improve their ability to extract global information. Jang et al. [51] introduced spatial attention into autoencoders and designed attention-based autoencoders to learn or adjust positional information in latent space. Plakias et al. [52] incorporated spatial attention into densely connected CNNs to enhance the model’s feature extraction capability, reducing the required amount of data and enabling the identification of bearings with different degrees of damage. A popular channel attention technique, SE-Net, has been widely adopted in mechanical diagnostics. Hao et al. [53] introduced SE attention to a multi-scale CNN for feature fusion and proposed a bearing fault diagnosis method. Yang et al. [54] designed a multi-attention approach that combines SE-Net and global attention, assigning reasonable weights to CNN feature maps to improve the diagnosis of aircraft engine faults.

Despite the numerous studies on attention mechanisms in fault diagnosis in recent years, most of them are based on channel attention and spatial attention. However, these mainstream attention mechanisms in fault diagnosis currently have some limitations. For example, channel attention weights different channels of vibration signals based on their relationships but overlooks the positional relationships of the signals in the time series. This leads to an inaccurate capture of the temporal variations of fault signals [22]. Self-attention mechanisms are more suitable for processing shorter sequences, while they may encounter issues, such as imbalanced information propagation or vanishing gradients when dealing with longer sequences [55]. Global attention disperses attention weights too widely, allocating excessive attention to irrelevant parts of vibration signals [24]. We noticed that local attention mechanisms in neural machine translation and speech recognition had shown significant improvements. Luong et al. [25] proposed the local attention mechanism, enabling neural machine translation models to better model the relationships between source and target languages, thus improving translation quality. Mirsamadi et al. [26] proposed using local attention to focus on specific regions of the speech signal that are more significant in terms of emotion. Therefore, we introduce the concept of local attention mechanism into the field of fault diagnosis, where local attention focuses on specific regions of input information, shifting from global attention to local attention. By adaptively highlighting features in local regions with weighted pooling, important contextual information is preserved, enabling better identification of fault features when dealing with long sequences, such as vibration signals. To the best of our knowledge, there has been no research on applying local attention mechanisms in fault diagnosis. In this paper, we propose a cross-sensor domain transfer diagnosis method that combines local attention, utilizing this novel weighted pooling strategy to focus on specific parts containing fault signals.

## 3. Problem Formulation

Transfer learning utilizes knowledge from a source domain to assist in establishing a predictive model for the target domain, thereby improving the accuracy and reliability of tasks in the target domain [56]. This paper aims to address the issue of cross-sensor domain transfer diagnosis for reciprocating pumps. The proposed model will be trained on labeled data from one position sensor and then transferred to unlabeled data from another position sensor. Thus, the source and target domains are defined as follows.

(1)Construct a source domain:

(1)$$D_{s}={\left\{\left(x_{i}^{s},y_{i}^{s}\right)\right\}}_{i=1}^{n_{s}}\;x_{i}^{s}\in X_{s},y_{i}^{s}\in Y_{s}$$
where $D_{s}$ represents the source domain, $x_{i}^{s}$ is the ith source domain sample, $X_{s}\in D_{s}$ is the union of all samples, $y_{i}^{s}$ represents the label for the ith source domain sample, $Y_{s}$ is the union of all different labels, and $n_{s}$ means the total number of source samples.

(2)Construct a target domain:

(2)$$D_{t}={\left\{\left(x_{i}^{t}\right)\right\}}_{i=1}^{n_{t}}\;x_{i}^{t}\in X_{t}$$where $D_{t}$ represents the target domain, $x_{i}^{t}$ is the ith target domain sample, $X_{t}\in D_{t}$ is the union of all samples, and $n_{t}$ means the total number of target samples.

(3)The source domain should provide enough diagnosis knowledge for the target domain, i.e., $y_{t}\subseteq y_{s}\subseteq y$
where $y_{s}$ and $y_{t}$ are label spaces in the source and target domains, respectively. We also denote the label space ξ=(1,2,3…k), which contains k, which represents the kinds of health states.

The vibration signal data from the source and target domains are collected from different positions on the reciprocating pump. As a result, these data exhibit significant distribution differences. As shown in Figure 1a, if we use an intelligent diagnostic model to learn features directly from these data, the learned features will also suffer from substantial distribution discrepancies. Therefore, we aim to extract transferable features from the source domain data to reduce the cross-domain differences. As shown in Figure 1b, we hope to build a model β(⋅), which can classify unlabeled samples x in the target domain.
(3)$$\hat{y}=\beta (x)$$
where $\hat{y}$ is the prediction. Thus, transfer learning is aimed to minimize the target risk $\varepsilon_{t}(\beta )$ using source data supervision.
(4)$$\varepsilon_{t}(\beta )=\underset{{}_{(x,y)\sim Q}}{\mathrm{Pr}}[\beta (x)\neq y]$$

## 4. The Proposed Method

This paper proposes a transfer diagnosis method integrating local attention for cross-sensor domain fault diagnosis of reciprocating pumps. The architecture of the proposed method is illustrated in Figure 2. Firstly, the collected data are input into the convolutional layers for feature extraction. The raw sensor data are mapped into a feature representation, and the feature dimensions are adjusted accordingly. Next, the local features are input into the module of the local attention mechanism, which is explained in Section 4.2. Subsequently, multi-layer domain adaptation is employed to reduce the distribution differences of the learned transferable features, the trained model is used to test samples in the target domain directly, which means that source and target domains share the same model and parameters. Finally, the trained model predicts the health status of unlabeled data samples in the target domain of reciprocating pumps. We provide the whole processing of transfer diagnosis method in Algorithm 1.
**Algorithm 1** Transfer diagnostic procedure
**①****Training:** **Input:** Labeled source domain $D_{s}={\left\{\left(x_{i}^{s},y_{i}^{s}\right)\right\}}_{i=1}^{n_{s}}$, unlabeled target domain $D_{t}={\left\{\left(x_{i}^{t}\right)\right\}}_{i=1}^{n_{t}}$, *max_epoch, batch_size.* **Output:** The trained Transfer diagnostic model β(⋅)  1: Initialize: Feature extractor $f_{\theta }$, domain classifier $D_{\theta }$  2: Pretrain $f_{\theta }$ using source domain data  3: **for** *epoch = 1* to *max_epoch* **do**  4:    **for** *batch_size* $x_{s}\in D_{s}$, $x_{t}\in D_{t}$ **do**  5:     Conduct Transfer diagnostic model training  6:     Update $f_{\theta }$ using $x_{s}\;$to minimize source task loss  7:     Update $D_{\theta }\;$using $x_{s}$ and $x_{t}\;$to maximize domain classification loss  8:   end for  9: end for**②****Testing:** Fed the testing target domain samples $\beta \left(\cdot \right)$ for the fault diagnosis.

### 4.1. Model Architecture

As shown in Figure 2, the model backbone consists of four Conv layers, where each Conv layer includes a 1D convolutional layer, a 1D batch normalization (BN) layer, and a ReLU activation function. Additionally, the first combination comes with the local attention module, and the fourth combination comes with a 1D adaptive max-pooling layer to realize the adaptation of the input length.

The convolutional output is then flattened and passed through a fully connected (FC) layer; in addition, the dropout technique is employed to reduce overfitting. The detailed description of all parameters is presented in tabular form in Table 1.

### 4.2. Local Attention Module

Inspired by the attention mechanism in neural machine translation [25], we introduce a local attention mechanism in our model. The ability of a model to automatically divert more attention to the most critical features is known as the local attention mechanism, which significantly improves the efficiency and accuracy of models in learning complex information [57]. Essentially, the attention mechanism is an adaptive weighting operation on the input. To understand the local attention mechanism, it is necessary to clarify that the convolution operation mentioned in the text extracts spatial or temporal features from the input data, and convolution is a global operation that scans the entire input signal (e.g., an image). However, this convolution operation ignores the differences between different regions when processing certain time-series signals (e.g., speech, video, vibration signals), which affects the expressive power of the neural network model. Therefore, we solve this problem by adding a local attention module to the diagnostic model.

The core idea of the local attention mechanism is to assign different weights to the data within local regions to extract valuable information better [58], specifically targeting the feature extraction process in convolutional neural network (CNN) models. In CNN, the convolution operation extracts spatial or temporal features from the input data, but they usually average over the entire input signal, ignoring the differences between different regions. To address this problem, we employ a local attention mechanism to enhance the feature extraction capability of the model within local regions. As shown in Figure 3, the local attention mechanism selectively focuses on important feature information, which is subsequently fed into the next layer of the model for classification. Algorithm 2 describes a specific implementation of the local attention mechanism. The proposed solution can be formulated as follows:
**Algorithm 2** Local Attention Mechanism
**Input:** *x* (input tensor)**Output:** Adjusted *x* with local attention mechanism*  *1. **function** Local Attention(*x*):*  *2.   Initialize input parameters: *in_channel, kernel_size**  *3.   ① Initialize convolution layer:*  *4.   conv(*x*) ← Convld(*in_channel, out_channel, kernel_size,**        padding = (kernel_size − 1)//2*)*  *5.   **return** *w, x**  *6.   ② Initialize softmax activation function:*  *7.   softmax ← *SoftMax*(⋅)*  *8.   Calculate attention weights:*  *9.   *weights* ← softmax(*w* × *x*)*  *10.   ③ Apply attention:*  *11.   *x* ← *x* × *weights**  *12.   **return** *x**  *13. **end function**

The local attention module begins by taking the signals on each channel of the convolutional layer as the input tensor X. A 1D convolutional layer is used to calculate the weights for each time step. For an input signal $X\in R^{C\times T}$ of length T, the output of the convolutional layer is defined as:(5)$$Z_{i,j}=\sum_{k=i-1}^{i+1}w_{k}\cdot X_{j+k}$$

$w_{k}$ represents the kth elements in the convolutional kernel, which are typically learned automatically through gradient backpropagation. $X_{j+k}$ denotes the input signal at the time step j+k. The local attention mechanism calculates the weight at each time step, influenced by the size of the convolutional kernel. Therefore, the range of k values is determined by the kernel_size, which falls within the interval:(6)$$k\in \left[-\left\lfloor kernel\_size/2\right\rfloor ,\left\lceil kernel\_size/2\right\rceil -1\right]$$

$\left\lfloor kernel\_size/2\right\rfloor$ represents the maximum integer not exceeding kernel_size/2, and $\left\lceil kernel\_size/2\right\rceil \;$represents the minimum integer not less than kernel_size/2. For example, setting the convolutional layer with kernel_size=3 and stride=1, the output $Z_{i,j}$ depends solely on $X_{j-1},X_{j},X_{j+1}$, which indicates the localized information. We apply a *SoftMax* activation function to the output $Z\in R^{C\times T}$of the convolutional layer for computation:(7)$$\alpha_{j}=\frac{\exp\left(Z_{i,j}\right)}{\sum_{k=1}^{T}\exp\left(Z_{i,k}\right)}$$

$\alpha_{j}$ represents the weight coefficient corresponding to the jth time step, obtained by normalizing the *SoftMax* function mentioned above. Finally, we calculate the weighted sum of the input signals based on the weights corresponding to each time step, and the result is considered as the output of local attention, denoted as $Att(X)\in R^{C\times 1}$:(8)$$Att(X)=\sum_{j=1}^{T}\alpha_{j}\cdot X_{j}$$

By introducing the local attention mechanism, the more capable the CNN model becomes of extracting effective feature representations from local details, the better it is capturing and learning the features of the input data. Consequently, this mechanism effectively enhances the performance of the fault diagnosis model.

## 5. Case Study

### 5.1. Dataset Description

Experiment Description

In this case study, the proposed method is used to diagnose the reciprocating pumps’ faults. The reciprocating pump model is CDWL25-0.4, The rated power is 30 kW, and the rated speed of the drive motor is 1460 r/min. The INV3065N2 Multi-function Dynamic Signal Test System and the Piezoelectric accelerometer INV982X were employed for vibration signal acquisition, and the sampling frequency of 10 kHz is used in the experiment. The signal collection is completed in Chongqing Pump Industry.

As shown in Figure 4, the object of diagnosis is a vertical reciprocating pump, whose drive mechanism makes a reciprocating motion in the vertical direction. Six vibration sensors are arranged vertically on the pump head and machine foot of the reciprocating pump, respectively, and the vibration signal data collected by the sensors are used to evaluate the effectiveness and feasibility of the proposed method in cross-sensor domain migration. Table 2 shows the information for each measurement point.

2.Operating conditions

It is worth noting that the faults used in the experiments were naturally occurring failures in the reciprocating pumps during their operational processes, rather than artificially induced. Faulty components from the reciprocating pumps that experienced failures were utilized for the experiments, and corresponding data were collected. As depicted in Table 3, there are five types of faults: valve sealing surface compression, valve sealing surface erosion, valve sealing surface indentation, check valve guide fracture, and valve assembly corrosion. In addition to the normal state, a total of six operating conditions were considered. Table 4 shows fault information of the data collected by the sensors in the six operating conditions.

Operating Condition 1: Normal StateOperating Condition 2: Valve Seat Compression Injury

During the moment when the one-way valve in the pump closes, and solid particles are present in the fluid, some of these particles may not be discharged and can become trapped and compressed between the sealing surfaces. This produces extremely high localized forces, leading to compression injury on the specific sealing surface, as shown in Figure 5a.

Operating Condition 3: Valve Seat Erosion

The clearance between the one-way valve and the sealing surface of the valve seat is minimal, representing the narrowest section in the entire fluid flow passage. The fluid velocity is high. If there is poor sealing, as illustrated in Figure 5b, the high-speed fluid will cause erosion on the sealing surface. This is a common form of damage to the one-way valve in the pump.

Operating Condition 4: Valve Seat Depression

The valve seat remains stationary during operation and endures the impact load from the one-way valve and erosion from the fluid. Due to the wide sealing surface of the valve seat, a portion of it experiences impact from the metal sealing surface, while another portion experiences impact from the rubber seal. Due to the different impact forces and materials, certain sections of the sealing surface may depress, forming two conical surfaces. Simultaneously, the roughness of the sealing surface increases, as shown Figure 5c, creating local grooves that lead to a loss in sealing capability and subsequent failure.

Operating Condition 5: Guiding Failure of Check Valve

The guiding mechanism of the one-way valve ensures its linear motion and maintains the motion along the centerline of the bore in the valve seat. Otherwise, it would cause delayed closure or incomplete closure of the one-way valve, resulting in rapid erosion and failure of the sealing surfaces. Thus, the guiding component is crucial for the proper functioning of the one-way valve. Typically, the guiding mechanism of the one-way valve consists of three or four lobes, as depicted in Figure 5d. If any of these lobes fracture or suffer severe wear, they will fail to perform their guiding function effectively.

Operating Condition 6: Corrosion of Valve Assembly

The one-way valve in the pump is a component directly exposed to the conveyed medium. When the medium is corrosive, such as when it contains sulfuric acid, hydrochloric acid, nitric acid, and other corrosive substances, it can cause corrosion of the one-way valve, as shown in Figure 5e. When the sealing surfaces or guiding components are corroded and fail, the reciprocating pump will also fail to operate correctly.

### 5.2. Transfer Task Description

In general, the position of the pump head is more sensitive to fault identification than the machine foot position. However, the valve seat position is more convenient for installing vibration sensors. To validate the cross-sensor domain transferability between the pump head and machine foot positions, two transfer tasks are designed:(1)Task 1: As shown in Table 5, Task 1 consists of nine cross-sensor domain fault diagnosis experiments, namely A→D, A→E, A→F, B→D, B→E, B→F, C→D, C→E, and C→F. In each fault diagnosis experiment, the part before the arrow represents the source domain, and the part after the arrow represents the target domain. The aim is to determine the effectiveness of transferring from the fault-critical sensor position (pump head position) to the sensor position that is easier to install (machine foot position).(2)Task 2: Task 2 consists of nine cross-sensor domain fault diagnosis experiments, evaluating the transferability from the machine foot position to the pump head position. The goal is to demonstrate the possibility of transferring signals from a location with less healthy information to a location with more healthy information.

### 5.3. Data Preprocessing and Splitting

In both the source and target domains, there are 1000 samples for each class. To improve the diagnostic accuracy, a sliding sampling technique is employed to partition the original data, expanding the fault samples. There is an overlap between adjacent samples, and each sample contains 2048 data points to capture sufficient fault information. The data in the target domain are used for the training process to achieve transfer learning and serves as the test set. Therefore, as shown in Figure 6, 80% of the total samples is used as the training set, while the remaining 20% is used as the test set, both in the source and target domains. Furthermore, to minimize additional computations and the influence of domain-specific knowledge, this study directly uses the original vibration samples as inputs to the fault diagnosis model.

### 5.4. Training Details

We implement all transfer tasks methods in Pytorch and put them into a unified code framework. Each model is trained for 100 epochs, and model training and test processes are alternated during the training procedure. We adapt the minibatch Adam optimizer, and the batch size is equal to 128. The “step” strategy in Pytorch is used as the learning rate annealing method, and the initial learning rate is 0.001 with a decay (multiplied by 0.1) in epochs 50 and 75.

The program is coded in Pytorch 2.0 and runs on GeForce RTX 4070 with 32G RAM in Windows 11 platform.

## 6. Results and Discussion

### 6.1. Comparative Methods

To further evaluate the effectiveness and superiority of the proposed diagnostic model, several well-known and state-of-the-art attention models are compared:

CNN: Convolutional Neural Network is a famous deep learning approach that adaptively extracts fault features and achieves fault classification. It is used as a baseline to highlight the diagnostic capabilities of other transfer learning models.

SENet: Squeeze-and-Excitation Network [53] is a network architecture that introduces a channel attention mechanism to enhance the performance of CNN. It allows the model to focus on relevant feature channels related to the target while attenuating irrelevant feature channels, thereby improving the model’s performance.

ECANet: Efficient Channel Attention [59] is an extremely lightweight channel attention module that reduces the computational burden of attention modules through techniques, such as low-rank tensor decomposition and low-rank convolution.

GANet: Global Attention [24] enhances the model’s ability to model global information. It involves global pooling operations on the entire input sequence or feature maps to capture global contextual information.

### 6.2. Experimental Results

In order to ensure a fair comparison, the backbone network of each model remains consistent. The accuracy of the two cross-sensor domain tasks is presented in Table 6 and Table 7, respectively.

The conclusions drawn from these two tables are as follows:Compared to CNN, other models demonstrate higher accuracy in the target domain. This suggests that attention mechanisms can effectively enhance the accuracy of fault diagnosis. In addition, the proposed method achieves higher accuracy compared to other attention mechanism models, indicating that the local attention mechanism is well-suited for vibration signal fault diagnosis.The overall accuracy of Task 1 is higher than that of Task 2. This discrepancy arises from the heightened sensitivity of the pump head position to the fault identification, highlighting the ability to extract more distinct fault information from the sensors located at the pump head. Hence, training that uses pump head data as the source domain is a viable approach.The results show that the B position in the middle of the pump head of the reciprocating pump to the E position in the middle of the machine foot shows the highest accuracy compared to other scenarios. This is due to the proximity of measurement points B and E to the drive mechanism of the reciprocating pump, enabling them to most directly reflect abnormal vibrations. Our results corroborate this observation.

To visually showcase the advantages of the proposed method, we utilize t-Distributed Stochastic Neighbor Embedding (t-SNE) [60] to map the learned high-dimensional features into a two-dimensional space. Taking the experimental results of Task 1, from B to E, as an example, as shown in Figure 7, it can be observed from Figure 8a that the transferable feature information learned by the CNN model has significant overlap, resulting in low prediction accuracy. Figure 7b–d demonstrates that SENet, ECANet, and GANet improve upon the transferable features learned by the CNN model, particularly GANet, indicating that incorporating attention mechanisms effectively enhances the model’s feature learning. However, some feature overlap is still due to the small inter-class distance in the target domain. Figure 7e demonstrates that the proposed local attention mechanism significantly increases the inter-class distance of the transferable features extracted. These results clearly explain why the proposed method outperforms others in terms of prediction accuracy.

Figure 8 presents the confusion matrices obtained by the diagnostic models in six transfer experiments of Task 1. In Figure 8e, the proposed method achieves high diagnostic accuracy across all operating conditions with minimal misclassifications. Furthermore, we observe that almost all models can accurately diagnosis operating condition 1, which corresponds to the normal state. The normal state may possess the most similar features, making it easier to align distributions. The differences in diagnostic capability primarily concentrate on operating conditions 2, 3, and 6. The experimental results in the figure further validate the effectiveness of the proposed method.

## 7. Conclusions

In this paper, a cross-sensor transfer diagnosis method is proposed, which utilizes the sharing of information collected by sensors between different locations of the machine to achieve a more accurate and comprehensive fault diagnosis of the machine. The local attention mechanism is embedded in the proposed method to enhance the model’s ability to perceive the critical part of the fault signal. Finally, the proposed method is validated by applications of vibration signal data of reciprocating pumps. The experimental results show that the method achieves excellent results in several cross-sensor migration tasks. Thus, the method proposed in this paper can effectively solve the cross-sensor domain fault diagnosis problem of reciprocating pumps due to the structural conditions and operational environment. In the future, we will investigate more reciprocating pump fault classes and more complex sensor locations, and further applications and explorations will be conducted based on the proposed methods.

## Figures and Tables

**Figure 1 sensors-23-07432-f001:**
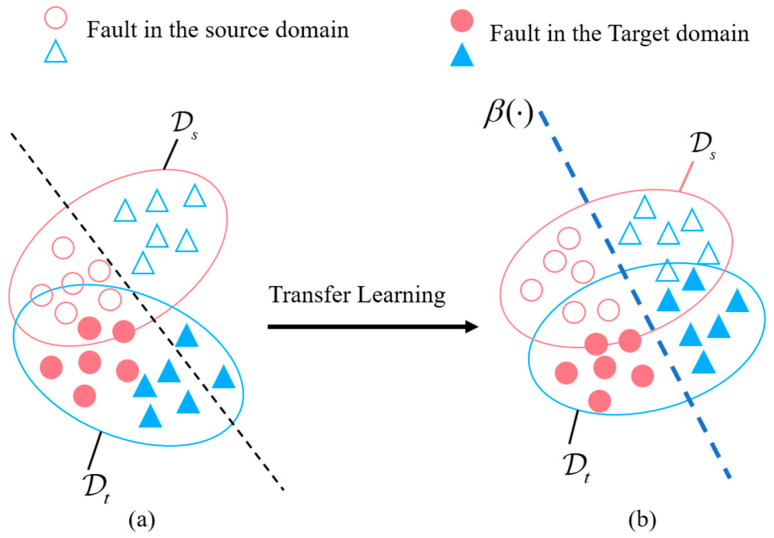
Intelligent fault diagnosis by feature-based transfer learning: (**a**) without transfer learning, and (**b**) with transfer learning.

**Figure 2 sensors-23-07432-f002:**
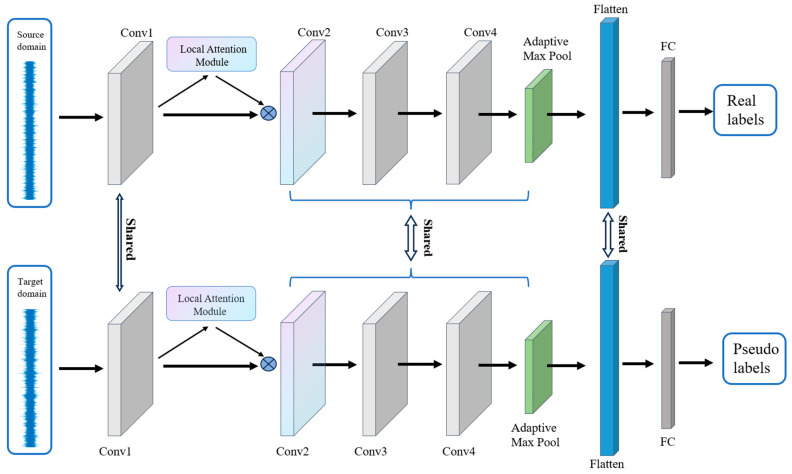
Architecture of the proposed model.

**Figure 3 sensors-23-07432-f003:**
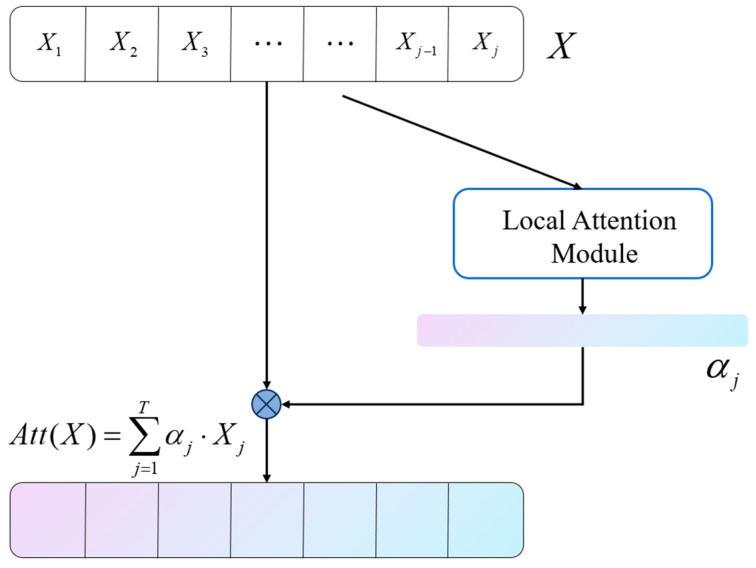
Flow chart of local attention.

**Figure 4 sensors-23-07432-f004:**
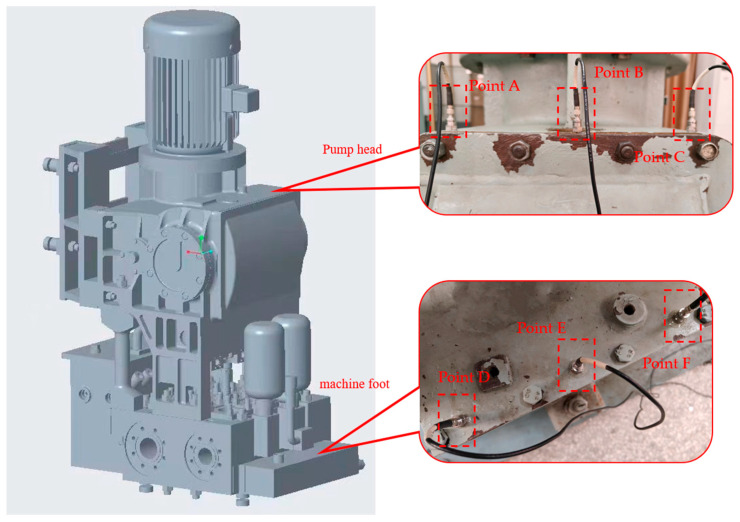
Reciprocating pump’s experimental setup.

**Figure 5 sensors-23-07432-f005:**
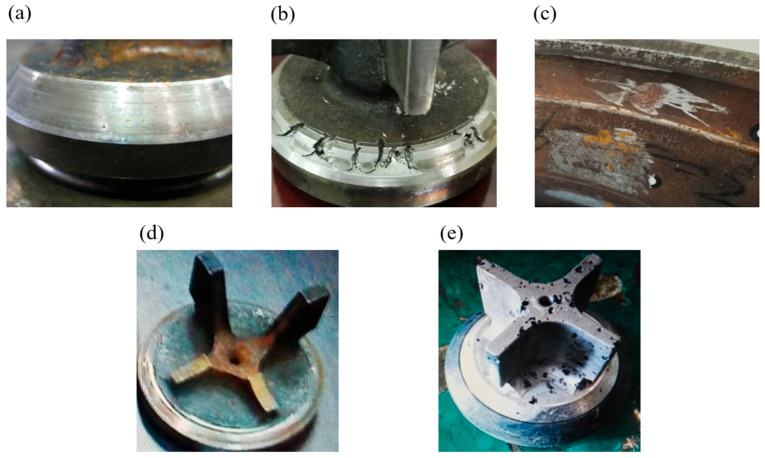
Types of reciprocating pump faults: (**a**) Valve Seat Compression Injury, (**b**) Valve Seat Erosion, (**c**) Valve Seat Depression, (**d**) Guiding Failure of Check Valve, (**e**) Corrosion of Valve Assembly.

**Figure 6 sensors-23-07432-f006:**
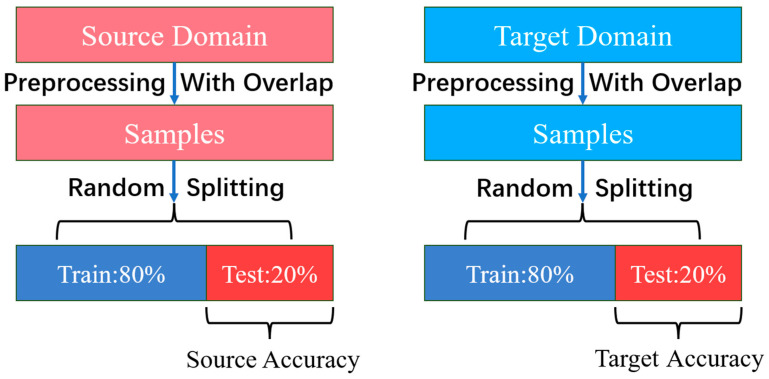
Data splitting for source and target Domain.

**Figure 7 sensors-23-07432-f007:**
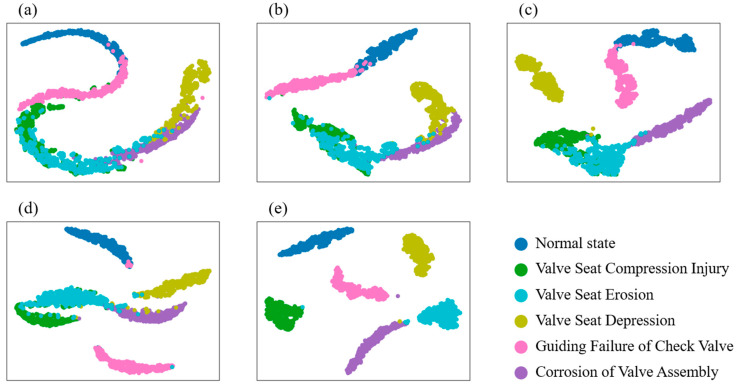
t-SNE visualization of B→E in Task 1: (**a**) CNN, (**b**) SENet, (**c**) ECANet, (**d**) GANet, (**e**) proposed method.

**Figure 8 sensors-23-07432-f008:**
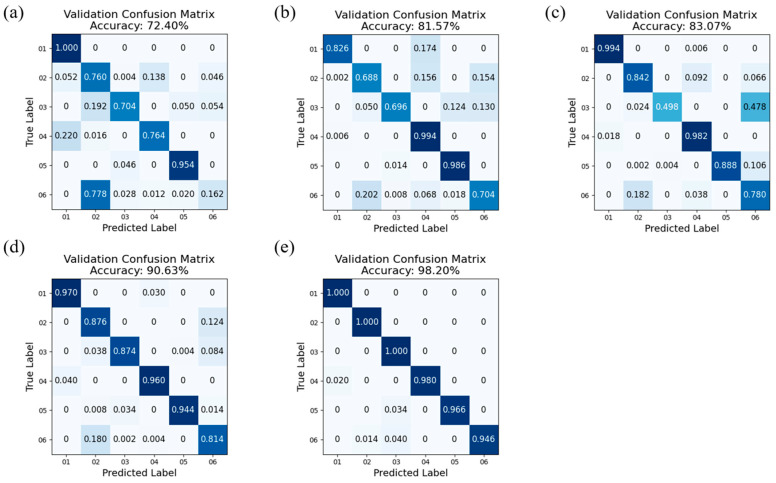
Confusion matrix of B→E in Task 1: (**a**) CNN, (**b**) SENet, (**c**) ECANet, (**d**) GANet, (**e**) proposed method.

**Table 1 sensors-23-07432-t001:** Parameters of the backbone.

Layer	Parameters	Values
Conv1	out_channels	8
	kernel_size	5
	stride	1
	batchnorm_size	8
Local Attention module	in_channels	8
	out_channels	8
Conv2	out_channels	16
	kernel_size	3
	stride	1
	batchnorm_size	16
Conv3	out_channels	32
	kernel_size	3
	stride	1
	batchnorm_size	32
Conv4	out_channels	64
	kernel_size	3
	stride	1
	batchnorm_size	64
Adaptive Max Polling	output_size	4
Flatten	-	-
FC	output_features	256
	pdrop	0.5

**Table 2 sensors-23-07432-t002:** Measurement point number and name.

Point Label	Point Name	Point Label	Point Name
A	Vertical direction of pump head 1	D	Vertical direction of machine foot 1
B	Vertical direction of pump head 2	E	Vertical direction of machine foot 2
C	Vertical direction of pump head 3	F	Vertical direction of machine foot 3

**Table 3 sensors-23-07432-t003:** Dataset description.

Condition Label	Operating Conditions	Number of Samples	Length of Samples
1	Normal state	6 × 1000	2048
2	Valve Seat Compression Injury	6 × 1000	2048
3	Valve Seat Erosion	6 × 1000	2048
4	Valve Seat Depression	6 × 1000	2048
5	Guiding Failure of Check Valve	6 × 1000	2048

**Table 4 sensors-23-07432-t004:** Label and Names of Sensor-Collected Data for Each Operating Condition.

Data Label	Data Name	Data Label	Data Name	Data Label	Data Name
1-A	Sensor A in Condition 1	2-A	Sensor A in Condition 2	3-A	Sensor A in Condition 3
1-B	Sensor B in Condition 1	2-B	Sensor B in Condition 2	3-B	Sensor B in Condition 3
1-C	Sensor C in Condition 1	2-C	Sensor C in Condition 2	3-C	Sensor C in Condition 3
1-D	Sensor D in Condition 1	2-D	Sensor D in Condition 2	3-D	Sensor D in Condition 3
1-E	Sensor E in Condition 1	2-E	Sensor E in Condition 2	3-E	Sensor E in Condition 3
1-F	Sensor F in Condition 1	2-F	Sensor F in Condition 2	3-F	Sensor F in Condition 3
4-A	Sensor A in Condition 4	5-A	Sensor A in Condition 5	6-A	Sensor A in Condition 6
4-B	Sensor B in Condition 4	5-B	Sensor B in Condition 5	6-B	Sensor B in Condition 6
4-C	Sensor C in Condition 4	5-C	Sensor C in Condition 5	6-C	Sensor C in Condition 6
4-D	Sensor D in Condition 4	5-D	Sensor D in Condition 5	6-D	Sensor D in Condition 6
4-E	Sensor E in Condition 4	5-E	Sensor E in Condition 5	6-E	Sensor E in Condition 6
4-F	Sensor F in Condition 4	5-F	Sensor F in Condition 5	6-F	Sensor F in Condition 6

**Table 5 sensors-23-07432-t005:** Transfer tasks of dataset.

Task	Source Domain	Target Domain
1	A	D, E, F
	B	D, E, F
	C	D, E, F
2	D	A, B, C
	E	A, B, C
	F	A, B, C

**Table 6 sensors-23-07432-t006:** Experimental results on the transfer diagnosis Task 1.

Methods	The Accuracy (%) of Cross-Sensor Transfer Diagnosis Task 1									AVG
	A→D	A→E	A→F	B→D	B→E	B→F	C→D	C→E	C→F	
CNN	62.96	54.83	48.21	49.12	72.40	69.29	63.38	62.61	70.09	61.43
SENet	80.82	72.14	64.33	65.09	81.57	79.23	78.56	80.83	79.68	75.81
ECANet	81.11	74.63	67.70	72.82	83.07	82.24	79.83	81.53	82.32	78.36
GANet	88.22	86.34	78.33	79.89	90.63	88.96	85.72	86.68	89.36	86.01
Proposed Method	94.83	92.76	88.52	89.03	98.20	96.92	95.86	95.07	96.12	94.15

**Table 7 sensors-23-07432-t007:** Experimental results on the transfer diagnosis Task 2.

Methods	The Accuracy (%) of Cross-Sensor Transfer Diagnosis Task 2									AVG
	D→A	D→B	D→C	E→A	E→B	E→C	F→A	F→B	F→C	
CNN	58.27	50.68	40.17	44.15	68.77	64.40	60.01	59.05	65.79	56.81
SENet	80.48	71.39	62.86	63.28	80.75	79.01	78.21	78.42	78.73	74.79
ECANet	77.54	71.41	63.86	68.65	79.16	78.74	75.10	77.09	77.85	74.38
GANet	85.21	76.80	73.04	74.83	87.23	83.27	83.44	85.13	87.85	81.87
Proposed Method	90.63	89.88	87.73	87.32	94.06	92.71	92.27	90.28	92.30	90.80

## Data Availability

Data can be made available upon reasonable request.

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
