# Peer review of "A Novel Cross-Sensor Transfer Diagnosis Method with Local Attention Mechanism: Applied in a Reciprocating Pump"

_sensors, 2023, doi:10.3390/s23177432_

Round 1

Reviewer 1 Report

In this paper, a cross-sensor transfer diagnosis method for reciprocating pump diagnosis is proposed. To bolster the fault diagnosis capability of the transfer learning mode, a local attention mechanism is introduced in this work.

While the authors have presented their idea clearly and the paper is well written, it is essential to acknowledge that the study's basic framework exhibits significant limitations. These limitations reduce the paper's overall attractiveness for publication; thus, the reviewer cannot accept this paper in its current form.

Here are some comments for the authors to consider, which may help in improving this study.

Comment-1: An accelerometer measures the rate of change of velocity from zero-to-peak, with a frequency response generated between the range of approximately 0 to 14000 Hz. The acquired vibration signal consists of displacement (0 ~ 200 Hz), velocity (10 ~ 2000 Hz), and acceleration (5 ~ 20 kHz). As these quantities possess both magnitude and direction, they are considered vectors.

Based on this characteristic, the vibration signals acquired by the accelerometer are directional. Hence, before obtaining the vibration signals from a mechanical specimen, regardless of its structural complexity, the first step is to determine the wave propagation path. Subsequently, the accelerometers are installed in alignment with the wave's direction to effectively capture the machine defect frequencies. In the industry, the diagnosis of specimens, irrespective of their structural complexity, typically involves either a minimum of one accelerometer installed along the wave propagation path or a maximum of three accelerometers installed in the x, y, and z directions of the wave propagation.

Utilizing accelerometers placed outside the wave propagation path would be less effective and fail to collect accurate defect signatures from the specimen. All the aforementioned considerations align with the ISO standards ISO 8528-1 to 8528-10, ISO 7919, and ISO 10816.

The data acquisition setup presented in this study utilized 6 accelerometers to acquire data from the reciprocating pump, without adhering to the ISO standards. Furthermore, the attachment of an abundant number of accelerometers to the specimen was done without determining the wave propagation path. However, it's worth noting that even a single accelerometer, when appropriately placed in the direction of the wave propagation, can effectively diagnose valve defects in reciprocating pumps.

The author justified the use of multiple accelerometers in line no. 46 by stating, 'Consequently, sensors are difficult to install in the most sensitive pump head positions that reflect faults.' However, it's important to consider that we are currently in 2023, and the market offers numerous waveguides that are significantly more cost-effective than accelerometers. Many industries utilize waveguides for data acquisition in complex mechanical structures, where relevant. Therefore, the rationale presented by the author for using multiple accelerometers is deemed invalid.

Furthermore, employing six sensors solely for diagnosing one mechanical specimen is not economically viable. Consequently, the proposed method may be less appealing to the industry, which consistently seeks cost-effective solutions.

Considering the points discussed above, the reviewer cannot accept this paper with its current framework.

The comments below can be helpful in improving the clarity of this work.

Comment-2: Why did the author choose vibration-based non-destructive testing over orbit analysis, deflection shape analysis, and acoustic emission analysis? This point requires a clear explanation in the introduction section.

Comment-3: The local attention mechanism calculates the weights alpha (α), as shown in Figure 3. Could you provide more information about alpha? Specifically, is the time step varying or fixed? Additionally, what is the length of the time steps considered in this study?

Comment-4: The local attenuation mechanism appears to prioritize the time step at which fault-specific frequencies can be detected. If this statement holds true, would it be possible for the authors to present the frequency spectrum of the vibration signal at a high alpha value?

Comment-5: The paper lacks a proper transfer learning network architecture. It is essential to include a comprehensive network architecture, along with all parameters, presented in tabular form, accompanied by a detailed explanation.

Comment-6: The author should include a picture of the actual test setup, along with schematics highlighting the position of each accelerometer attached to it. Additionally, in the experimental description, please provide more information about the power rating, type of pump, and type of accelerometer used in this study.

Comment-7: Could you please provide an explanation for why cases B->E and E->B exhibit the highest accuracy compared to the other cases presented in Table 3?

Comment-8 Why CNN is bold-faced in Tables 5 and 6.

Author Response

Response to Reviewer 1 Comments

Reviewer 2 Report

1.The fourth part of the manuscript focuses on the local attention mechanism and lacks the pseudo-code of the whole model.

2.Figure 5 in the manuscript is not clear enough, please modify to show the positions of the 3 sensors.

The manuscript needs extensive revision for language and grammar.

Author Response

Response to Reviewer 2 Comments

Reviewer 3 Report

1. In the Section 1 Introduction, Many research attempts have been made by several researchers on addressing cross-domain machinery condition monitoring problems. But the authors seem to ignore that part. Some newest works are not included, such as A dual drift compensation framework based on subspace learning and a cross-domain adaptive extreme learning machine for gas sensors (https://doi.org/10.1016/j.knosys.2022.110024)

2. Figure 1-(a) in line 232 and Fig.1b in line 226 are inconsistent. Also, the content in Figure 1 is easy to confuse the reader. For the same type of health states collected by different sensors, markers with the same shape and different colors should be used to distinguish them.

3. In Figure 7, whether the authors trained the model using 80% of the labeled samples in the target domain?

Minor revisions are needed.

Author Response

Response to Reviewer 3 Comments

Round 2

Reviewer 1 Report

The paper has undergone significant improvement through the revision process; nevertheless, some issues still require attention.

Comment-1: The explanations given in response to Comment-1 and Comment-7 should be appended to the end of section 6.2 (Experimental Results) in a separate paragraph. This placement will allow future researchers and industry professionals to reference those specific sensor locations for diagnosing faults in vertical reciprocating pumps.

Comment-2: In the conclusion section, there is a need to modify the sentence (538 to 539). This modification should entail the inclusion of specific details regarding the higher accuracy achieved by the authors using sensors B to E and E to B.

Comment-3: Regarding comment-4, the authors addressed future work. However, in the revised manuscript version, the reviewer is unable to locate this future direction within the "Future Work" presented in the conclusion section.

Comment-4: In the introduction of this paper, particularly within the paragraph where the author emphasizes the preference for vibration signals over other non-destructive testing techniques, the following papers can be briefly elucidated.

DOI-1: https://doi.org/10.3390/s22010179

DOI-2: 10.1109/ACCESS.2021.3124903

Comment-5: The heading of section 4.1 should be changed to "Model Architecture" instead of "Structure of Model."

Comment-6: Please review the revised version for grammatical issues. Additionally, in Section 5.1, kindly replace lines 357 to 362 with the grammatically correct rendition “The vibration signals were acquired using the INV3065N2 Multi-function Dynamic Signal Test System, and the Piezoelectric accelerometer INV982X was utilized for signal acquisition. Various operating conditions were configured for the experiment, with a sampling frequency of 10 kHz employed. The signal collection was conducted at Chongqing Pump Industry.”

Fine.

Author Response

Response to Reviewer 1 Comments (Round 2)

Dear Reviewers:

Thank you for your comments concerning our manuscript (Manuscript-ID: sensors-2537066). Those comments are all valuable and important to enhance the quality of our work. We have studied comments carefully and have made correction which we hope meet with approval. The modified parts in the revision are colored with red. The Point-to-Point responses are as follows:

Comment-1: The explanations given in response to Comment-1 and Comment-7 should be appended to the end of section 6.2 (Experimental Results) in a separate paragraph. This placement will allow future researchers and industry professionals to reference those specific sensor locations for diagnosing faults in vertical reciprocating pumps.

Response 1: Thank you very much for your comments. We are very sorry for not adequately explaining. We attach the explanations of Comment-1 and Comment-7 as separate paragraphs at the end of Section 6.2 (Experimental Results). In lines 511 to 515 of the manuscript: “The results show that the B position in the middle of the pump head of the re-ciprocating pump to the E position in the middle of the machine foot shows the highest accuracy compared to other scenarios. This is due to the proximity of measurement points B and E to the drive mechanism of the reciprocating pump, enabling them to most directly reflect abnormal vibrations. Our results corroborate this observation.”

Comment-2: In the conclusion section, there is a need to modify the sentence (538 to 539). This modification should entail the inclusion of specific details regarding the higher accuracy achieved by the authors using sensors B to E and E to B.

Response 2: Thank you very much for your comments. We followed the suggestion. We describe in the conclusion section the application of the method to reciprocating pumps with excellent results in several cross-sensor migration tasks. In addition, we add in the experimental results section of 6.2 specific details regarding the higher accuracy achieved by the authors using sensors B to E and E to B.

Comment-3: Regarding comment-4, the authors addressed future work. However, in the revised manuscript version, the reviewer is unable to locate this future direction within the "Future Work" presented in the conclusion section.

Response 3: Thank you very much for your comments. We have added this section within the "Future Work" presented in the conclusion section.

Comment-4: In the introduction of this paper, particularly within the paragraph where the author emphasizes the preference for vibration signals over other non-destructive testing techniques, the following papers can be briefly elucidated.

DOI-1: https://doi.org/10.3390/s22010179

DOI-2: 10.1109/ACCESS.2021.3124903

Response 4: Thank you very much for your suggestion. We followed the suggestion. We have added some recent research results on vibration signal-based fault diagnosis in lines 56-67 of the manuscript. These additional references enhance the credibility and depth of the paper, further supporting and complementing our research.

Comment-5: The heading of section 4.1 should be changed to "Model Architecture" instead of "Structure of Model."

Response 5: Thank you very much for your suggestions. We have changed the heading of Section 4.1 to "Model Architecture".

Comment-6: Please review the revised version for grammatical issues. Additionally, in Section 5.1, kindly replace lines 357 to 362 with the grammatically correct rendition “The vibration signals were acquired using the INV3065N2 Multi-function Dynamic Signal Test System, and the Piezoelectric accelerometer INV982X was utilized for signal acquisition. Various operating conditions were configured for the experiment, with a sampling frequency of 10 kHz employed. The signal collection was conducted at Chongqing Pump Industry.”

Response 6: Thank you very much for your comments. We have revised the English grammar of this manuscript in detail so that readers can express our work more clearly. We change the previous lines 357 to 362: “The INV3065N2 Multi-function Dynamic Signal Test System and the Piezoelectric accelerometer INV982X were employed for vibration signal acquisition, and the sampling frequency of 10 kHz is used in the experiment. The signal collection is completed in Chongqing Pump Industry.”
